David *et al. Genome Biology*　　(2022) 23:240

**RESEARCH**

# Retained introns in long RNA-seq reads are not reliably detected in sample-matched short reads

Julianne K. David[1,2,3†], Sean K. Maden[1,2,4†], Mary A. Wood[1,5,6], Reid F. Thompson[1,2,7,8,9*] and Abhinav Nellore[1,2,10*] 🔟

†Julianne K. David and Sean K. Maden contributed equally to this work.

*Correspondence:
thompsre@ohsu.edu;
anellore@gmail.com

[2] Department of Biomedical Engineering, Oregon Health & Science University, Portland, OR, USA
[9] Department of Radiation Medicine, Oregon Health & Science University, Portland, OR, USA
Full list of author information is available at the end of the article

## Abstract

**Background:** There is growing interest in retained introns in a variety of disease contexts including cancer and aging. Many software tools have been developed to detect retained introns from short RNA-seq reads, but reliable detection is complicated by overlapping genes and transcripts as well as the presence of unprocessed or partially processed RNAs.

**Results:** We compared introns detected by 8 tools using short RNA-seq reads with introns observed in long RNA-seq reads from the same biological specimens. We found significant disagreement among tools (Fleiss' $\kappa = 0.113$) such that 47.7% of all detected intron retentions were not called by more than one tool. We also observed poor performance of all tools, with none achieving an F1-score greater than 0.26, and qualitatively different behaviors between general-purpose alternative splicing detection tools and tools confined to retained intron detection.

**Conclusions:** Short-read tools detect intron retention with poor recall and precision, calling into question the completeness and validity of a large percentage of putatively retained introns called by commonly used methods.

**Keywords:** RNA-seq, Splicing, Intron retention

## Background

During RNA transcription, multiple spliceosomes may act on the same transcript in parallel to remove segments of sequence called introns and splice together flanking exons [1]. Most splicing occurs stochastically [2] during transcription [3–5], although up to 20% of splicing may occur after transcription and polyadenylation [5, 6] (Additional file 1: Fig. S1). Introns are spliced by U2 and U12 spliceosomes [7], primarily in the nucleus [8], though studies suggest that cytoplasmic splicing may also occur [9–12].

Intron retention (IR) is a form of alternative splicing where an anticipated intron remains after transcript processing is complete. IR occurs in up to 80% of protein-coding genes in humans [13] and may affect gene expression regulation [14–20] as well as

response to stress [21–23]. Transcripts containing introns may also be stably detained in the nucleus before undergoing delayed splicing ("intron detention," or ID), with implications for temporal gene expression [24]. In cancers, high levels of IR [25–27] can generate aberrant splicing products with known and potential biological consequences for gene expression and cell survival [28]. It is unclear how commonly IR may give rise to a protein product [29, 30], but novel peptides derived from transcripts with retained introns (RIs) are increasingly being studied in disease contexts such as cancer [31–35].

Despite its biological relevance, detection of IR from bulk RNA sequencing (RNA-seq) data remains challenging for two principal reasons: (1) a short RNA-seq read (e.g., from Illumina's HiSeq, NovaSeq, or MiSeq platforms) is almost never long enough to resolve a full intron or its context in a transcript, particularly in genome regions with multiple overlapping transcripts, and (2) RNA-seq data may contain intronic sequence from unprocessed or partially processed transcripts, DNA contamination, and non-messenger RNA such as circular RNAs (cRNAs) [4, 36], potentially yielding spurious IR calls, independent of read length.

Existing tools designed specifically for RI detection make simplifying assumptions to address the above issues. These tools include keep me around (KMA) [37], IntEREst [38], iREAD [39], superintronic [40], and IRFinder [13] and its most recent implementation as IRFinder-S [41]. Some mitigate challenge (1) by ignoring from consideration any intronic regions that overlap other features (KMA, IntEREst, and iREAD), leaving biological blindspots in RI detection [37–39]. Some attempt to mitigate challenge (2) by recommending that a user provides poly(A)-selected data as their input [13, 37, 39, 40], assuming that poly(A) selected data represents fully processed, mature RNA. However, poly(A) selection during library preparation has been shown not to remove all immature post-transcriptionally spliced RNA molecules, and intronic sequences are commonly found in poly(A)-selected RNA-sequencing data [42, 43]. Other tools (e.g., rMATS [44], MAJIQ [45], and SUPPA2 [46]) are designed to identify a broader range of alternative splicing events, not just RIs, and do not make this assumption. To clarify the quality of and best practices for RI detection, we performed tests on poly(A)-selected, sample-matched long- and short-read sequencing runs for two biological specimens, with processed long-read data providing a standard against which we evaluated short read-based RI detection across eight tools: five RI-specific tools (KMA, IntEREst, iREAD, superintronic, and IRFinder) and three general-purpose tools (rMATs, MAJIQ, and SUPPA2).

## Results

### Testing RI detection using sample-paired short- and deep long-read RNA-seq data

To generate a dataset to test RI detection, we identified two human biological specimens on the Sequence Read Archive (SRA) with RNA-seq data from both Illumina short-read and PacBio Iso-Seq RS II long-read platforms (Fig. 1). These were a human whole blood sample (HX1) [47] and a human induced pluripotent stem cell line sample (iPSC) [48], with, respectively, 46 and 27 Iso-Seq runs, 24.4 and 91.3 million aligned short reads, and 945 and 840 thousand aligned long reads (Additional file 1: Table S1). To confine attention to robustly represented loci, we identified a set of 1327 and 1203 target genes in HX1 and iPSC samples, respectively, each with ≥ 2 short reads per base median coverage across the

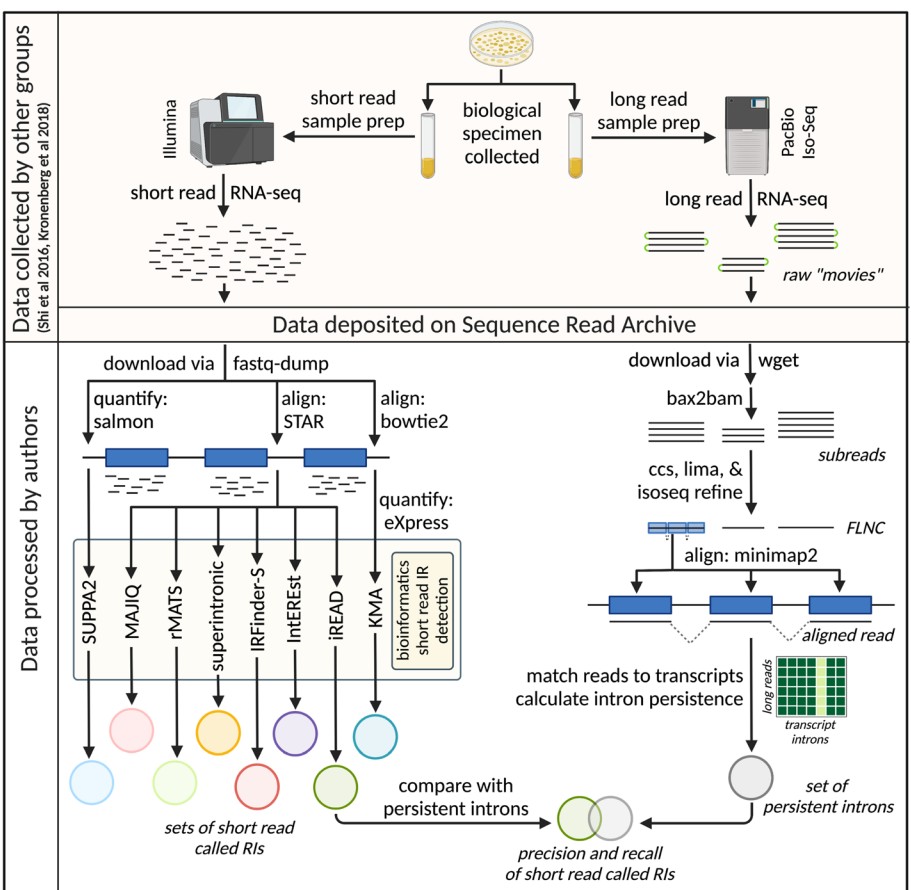

**Fig. 1** Overview of experimental plan. Long- and short-read RNA-seq data from the same biological specimen [47, 48] were downloaded from the SRA and subject to processing and analysis. Short reads (left path) were aligned and quantified according to the requirements of eight short read RI detection tools [37–41], and retained introns were called by each of these. The raw long Iso-Seq reads (right path) were processed to the stage of full-length non-concatemer (FLNC) reads, but left unclustered. After long reads were aligned to the reference genome, each aligned read was assigned to a best match transcript or discarded, and intron persistence was calculated. The called RI output of each short read detection tool was compared against the set of persistent introns identified in the long-read data (where $P_i \geq 0.1$). Created with BioRender.com

full gene length and $\geq 5$ full-length long reads assigned to at least one isoform of the gene (Additional file 1: Fig. S2).

We sought to quantify IR in each biological specimen using long-read data. To account for random splicing and sample contamination that may lead to noisy splicing patterns, we developed a novel variant of intron percent spliced in (PSI) [49]. For a given intron $i$ and transcript $t$, we defined persistence $P_{i,t}$ as

$$P_{i,t} = d_i \cdot \sum_{r \in M^t} \frac{R_{r,i} \cdot SF_{r,i} \cdot H_{r,i}}{|M^t|} , \tag{1}$$

where $r$ is a read among the set of all reads $M^t$ assigned as best matches to transcript $t$, information density $d_i$ is the proportion of $M^t$ covering intron $i$, the binary variable $R_{r,i}$ is 1 if and only if $r$ provides evidence for the retention of $i$, and the spliced fraction $SF_{r,i}$ and scaled Hamming similarity $H_{r,i}$ are defined in Section 5 (see Eqs. 3 and 4). In brief,

the intron persistence $P_{i,t}$ incorporates the extent and similarity of splicing across transcript reads, accounting for stochastic splicing initiation and progression (Additional file 1: Fig. S1). Note the information density, spliced fraction, and Hamming similarity modify standard transcript-specific PSI as captured by the $R_{r,i}$ factor. Finally, to address ambiguity in transcripts of origin in short-read data, we defined intron $i$'s persistence $P_i$ as the maximum persistence across all isoforms $T_i$ that contain $i$:

$$P_i = \max_{t \in T_i} P_{i,t} \, . \tag{2}$$

Going forward, we define a "persistent intron" as an intron for which $P_i \geq 0.1$.

Across the union of transcripts from both samples, a substantial majority (76.7%) of introns were fully spliced out ($P_{i,t} = 0$), and a small minority (0.13%) of introns were always unspliced within a transcript ($P_{i,t} = 1$) (Fig. 2a and Additional file 1: Fig. S3). These extreme values are in keeping with our qualitative understanding of splicing patterns; however, the range of intermediate persistence values appears to represent a spectrum with varying extents of inconsistent splicing across and between reads. While we tested short-read RI detection on a per-sample basis, we also compared intron persistence patterns between HX1 and iPSC samples and found significant similarity in splicing patterns across matched transcripts (Additional file 1: Figs. S3 and S4).

### Similarities of intron properties across short-read RI detection tool outputs

We compared RIs called by eight tools for short-read data (Table 1). While most introns were consistently spliced out, 80.8% (1072/1327) and 80.0% (963/1203) of target genes in HX1 and iPSC, respectively, had at least one RI identified in either short- or long-read data. Expression of called RIs varied substantially between tools in both HX1 (Fleiss' $\kappa = 0.145$) and iPSC (Fleiss' $\kappa = 0.068$), though we note that the outputs of RI-specific detection tools (IntEREst, superintronic, KMA, and IRFinder-S) are more correlated with each other than with the outputs of general-purpose alternative splicing

**Table 1** Short-read tools studied

| Tool | IRFinder-S [41] | super-intronic [40] | iREAD [39] | KMA [37] | IntEREst [38] | MAJIQ [45] | rMATS [44] | SUPPA2 [46] |
|---|---|---|---|---|---|---|---|---|
| Year | 2021 | 2020 | 2020 | 2015 | 2018 | 2016 | 2014 | 2018 |
| RI-specific | Yes | Yes | Yes | Yes | Yes | No | No | No |
| RI measure[a] | IRratio | log coverage | FPKM | TPM | FPKM or PSI | PSI | PSI | PSI |
| Language | C++ | R | Python | Python, R | R | Python | Python | Python |
| Host website | GitHub | GitHub | GitHub | GitHub | Bioconductor | Personal site | GitHub | GitHub |
| Sample data format | BAM or FASTQ | BAM | BAM | FASTQ | BAM | BAM | FASTQ | FASTQ |
| Reference format | GTF | GTF/GFF3 | BED | FASTA, GTF/GFF3 | GTF/GFF3 | GFF3 | GTF | GTF, FASTA |
| Intron definition | All introns | All introns | Independent introns[b] | Independent introns[b] | Independent introns[b] | All introns | All introns | All introns |

[a] See Section 5 for measure definitions

[b] Independent introns are intron regions not overlapping features from other transcript isoforms

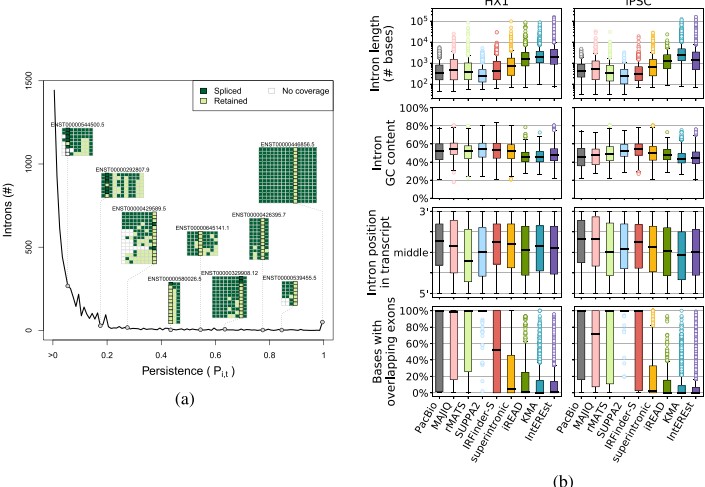

**Fig. 2** **a** Distribution of persistence $P_{i,t}$ and representative transcript examples for iPSC. The number of introns (*y*-axis) having a given persistence value (*x*-axis) is shown as a dark black line; note that a large number of introns with $P_i = 0$ are omitted from this analysis. Along the line, gray circles indicate the $P_i$ value corresponding to each of nine introns from representative transcript examples (each transcript is labeled by Ensembl ID, e.g., ENST00000446856.5). Read-level data is shown for each transcript as a colored matrix, where each row is a single long read assigned to the transcript and each column represents a given intron, and color indicates whether an intron is retained (light green), spliced out (dark green), or lacking sequence coverage (white) in a given read. **b** Distributions of properties of persistent and called RIs. Each panel contains a series of boxplots depicting the distribution of intron length (top, log-scale), % of bases in the intron that are G or C (2nd row), relative position in transcript (3rd row), and % of intron bases with overlapping annotated exons (bottom) for HX1 (left) and iPSC (right). The distribution of each of these features is shown for long-read persistent introns ("PacBio", gray) and RIs called by each of the eight short read tools: MAJIQ (light pink), rMATS (light green), SUPPA2 (light blue), IRFinder-S (red), superintronic (yellow), iREAD (green), KMA (blue), and IntEREst (purple)

detection tools (rMATS, MAJIQ, and SUPPA2) (Additional file 1: Fig. S5). Further, using circBase [50] to probe whether cRNA contamination may have affected RI detection, we identified only a small percent (< 7%) of called RIs that appeared to overlap intronic cRNAs (Additional file 1: Fig. S6).

We next examined the distributions of several intron properties (length, GC content, relative position in transcript, and annotated exon overlap) and their relationships with the set of RIs called by each short-read tool and their relative expression levels (Fig. 2b and Additional file 1: Fig. S7). Unsurprisingly, tools that exclude introns with overlapping genomic features (i.e. KMA, IntEREst, iREAD; Table 1) had exceedingly low overlap between exons and the RIs they reported. We also note that KMA and IntEREst called extremely long RIs (up to > 157 kilobases), compared to those called by other short-read tools or the persistent introns identified from long-read data (maximum 6275 and 5926 bases in HX1 and iPSC). We observed a slight overall 3′ bias among persistent introns from long-read data, as well as the set of RIs from several short-read tools (Fig. 2b), potentially reflecting the relatively shorter duration of exposure of 3′ introns to the cotranscriptional splicing machinery and/or implicit 3′ bias of the Clontech sample prep [51] used in both samples [47, 48]. Despite this slight 3′ tendency, there was no appreciable association between intron persistence and intron position in transcript

(Additional file 1: Fig. S8). Among all tools, IRFinder-S and MAJIQ called sets of RIs with characteristics most similar to persistent introns from long-read data (Fig. 2b).

### Precision and recall are poor across short-read RI detection tools

We tested performance (precision, recall, and F1-score) of RI detection by eight short-read tools, comparing sets of called RIs against persistent introns identified from long-read data (defined as $P_i \geq 0.1$). Overall tool performance was poor in most cases (Fig. 3a, Additional file 1: Table S2). Many persistent introns (41.0% and 27.7% in iPSC and HX1, respectively, Additional file 1: Fig. S9) were not called by any short-read tool, and the majority of called RIs were neither identified among persistent introns in long-read data nor consistently called between short-read tools (Fig. 3c and Additional file 1: Fig. S9). In HX1 and iPSC, respectively, 47.1% and 48.3% of called RIs were not called by more than one tool (47.7% overall). IRFinder-S had the best recall and F1-score, possibly due to the similarity between the properties of its called RIs and properties of persistent introns, though it is worth noting general-purpose alternative

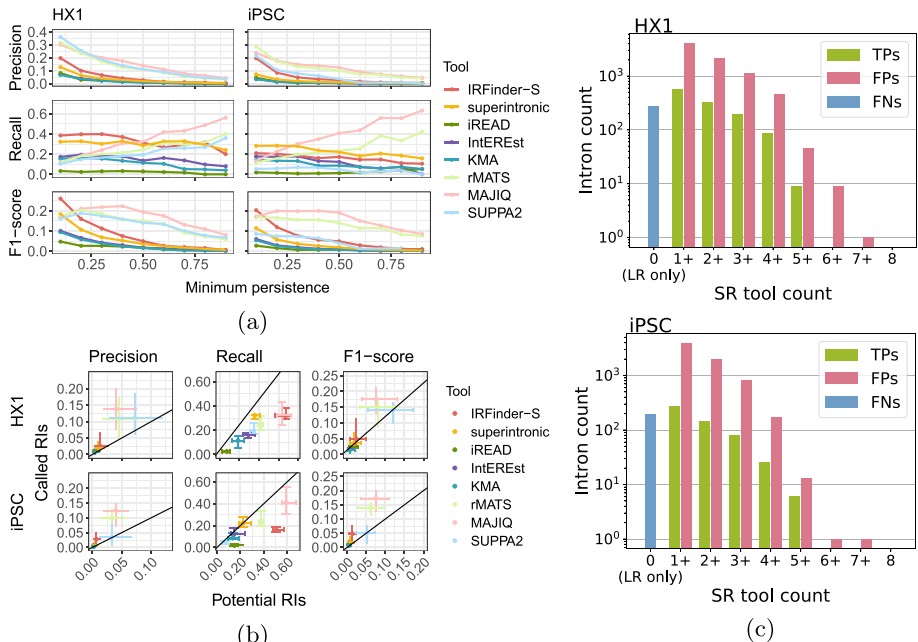

**Fig. 3** **a** Short-read tool performance across different thresholds of intron persistence. Each panel displays tool performance along the *y*-axis (measured by one of precision, recall, or F1-score as labeled) for a set of introns defined by the indicated threshold for intron persistence along the *x*-axis. Data for HX1 and iPSC are shown at left and right, respectively, with each tool's per-sample performance depicted in a different color (IRFinder-S [red], superintronic [yellow], iREAD [green], IntEREst [purple], and KMA [blue], MAJIQ [light pink], rMATS [light green], and SUPPA2 [light blue]). **b** Variation in short-read tool performance across intron persistence thresholds for potential vs. called RIs. Each panel displays tool performance as measured by precision (left), recall (middle), and F1-score (right) for HX1 (top) and iPSC (bottom) samples. The performances for each tool's potential RIs and called RIs are shown along the *x*- and *y*-axes, respectively, with centroid and whiskers denoting, respectively, the median and interquartile range of tool performance across intron persistence thresholds. Each tool's performance is depicted in a different color (color labels same as 3a.). Reference lines are shown with slope of 1. **c** Varying degrees of consensus of retained intron calls among short-read tools. Bar plots depict the number of true positive (green), false positive (pink), and false negative (blue) intron calls (y-axis) consistent across a specified number of short-read (SR) tools (x-axis). Upper and lower panels depict HX1 and iPSC data, respectively. LR denotes long-read data

splicing detection tools (rMATS, MAJIQ, and SUPPA2) had substantially higher precision than other tools. By contrast, iREAD demonstrated the lowest recall across all tools (Additional file 1: Fig. S10). Performance metrics for IntEREst and KMA were very similar across both samples (Fig. 3b).

To address sensitivity in persistent intron identification, we also considered short-read tool performance on subsets of long-read introns with increasing minimum thresholds of intron persistence ($P_i \geq 0.1$ to 0.9 in increments of 0.1). RI-specific tools (IRFinder-S, superintronic, iREAD, IntEREst, and KMA) exhibited qualitatively different behavior than general-purpose tools (rMATS, MAJIQ, and SUPPA2), with general-purpose tools performing better overall (Fig. 3a and Additional file 1: Fig. S11). MAJIQ in particular outperformed other tools across a range of persistence values, as measured by the F1-score. Among general-purpose tools, recall tended to increase with increasing persistence. Among RI-specific detection tools, overall performance remained poor across all levels of intron persistence, with uniformly worse precision, recall and F1-score as intron persistence increased. Further, IRFinder-S and superintronic were consistently best performers among RI-specific tools, albeit interchangeably depending on the sample, metric assessed, and intron persistence threshold. For instance, IRFinder-S demonstrated highest recall in HX1 at the lowest cutoff values ($P_i \geq 0.1$ to 0.4), while superintronic demonstrated higher recall across higher thresholds in HX1 and for all cutoffs in iPSC (Additional file 1: Table S2).

Finally, since each tool is capable of calling RIs with different levels of stringency, we evaluated tool performance on a raw set of all potential RIs (all expressed introns detected by that tool) vs. the corresponding subset of introns called as RIs by that tool. Rather than improving overall performance by retaining persistent RIs and removing false positive ones, stringency filters improved precision at the expense of recall, with typically slight corresponding improvements in F1-score across tools (Fig. 3b, Additional file 2).

### Short introns and introns that do not overlap exons are more reliably called

We next compared the distributions of seven intronic properties (length, GC content, position, exonic overlap, splice site motifs, U2- vs. U12-type spliceosomes, and uniformity of coverage by mapped reads) between the sets of true positive (TP), false positive (FP) and false negative (FN) RIs for each tool (defined as described in 5.18). Every tool except IRFinder-S had difficulty identifying shorter RIs (< 600 bases) (Fig. 4a, b). FPs tended to be longer than either TPs or FNs and were distributed more centrally within a transcript compared to persistent introns (both TPs and FNs) across all tools (Fig. 4a and Additional file 1: Fig. S12). As expected, the overwhelming majority of introns across all tools had canonical GT-AG splice motifs and splicing by the U2 spliceosome, while FNs showed increased frequencies of other motifs and spliceosome types relative to FPs and TPs (Additional file 1: Fig. S13).

We also probed how much distributional uniformity of mapped read coverage across an intron (coverage "flatness" [39, 41]) and incidence of overlapping exons differed among TPs, FPs, and FNs. Coverage of FPs and FNs was nonuniform, where coverage tended to decrease from 5′ to 3′ intron ends. Coverage of TPs was more

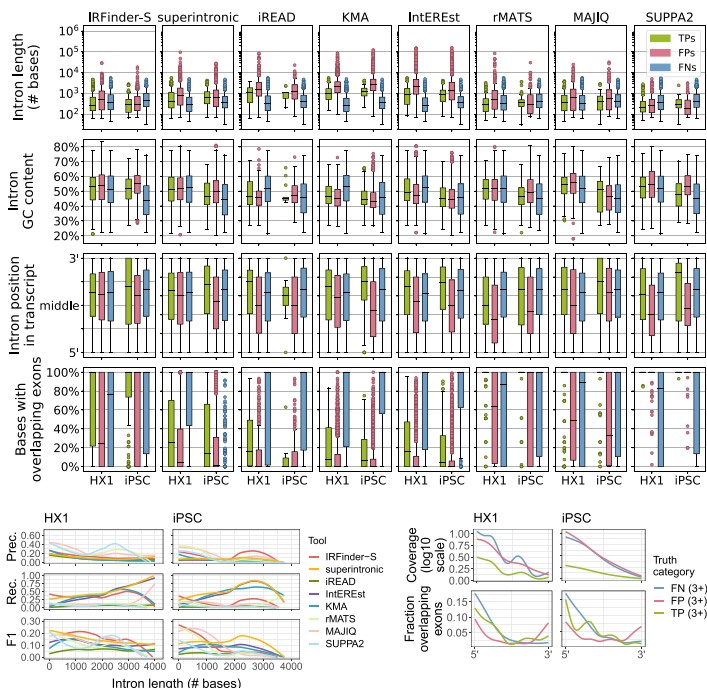

**Fig. 4 a** Distributions of properties of TP, FP, and FN RIs across short-read detection tools. Each panel displays the boxplot distributions of intron length (top, log scale), % of bases in the intron that are G or C (2nd row), relative position in transcript (3rd row), and % of intron bases with overlapping annotated exons (bottom) for the output from each of eight short-read tools (from left to right: IRFinder-S, superintronic, iREAD, KMA, IntEREst, rMATS, MAJIQ, and SUPPA2). *Y*-axes correspond to intron properties as labeled, with each boxplot along the *x*-axis corresponding to the TP (green, left boxes), FP (pink, middle boxes), and FN (blue, right boxes) calls for HX1 (left) and iPSC (right). **b** Short-read tool performance as a function of intron length. Each panel depicts the LOESS-smoothed precision (top), recall (middle), or F1-score (bottom) in either the HX1 (left) or iPSC (right) sample across overlapping, sliding window intron length ranges (Section 5). Smooths are grouped and colored by eight short-read tools (red = IRFinder-S, yellow = superintronic, green = iREAD, purple = IntEREst, blue = KMA, light pink = MAJIQ, light green = rMATS, light blue = SUPPA2). **c** Read coverage and exon overlap as a function of position within an intron. LOESS-smoothed short-read data (see Section 5) show the median log10-scaled coverage (top row, *y*-axes) and fractions of introns with overlapping exons (bottom row, *y*-axes) as a function of position (*x*-axis, 5′ → 3′ on positive strand) for HX1 (left column) and iPSC (right column). Introns were grouped by truth category membership for at least 3/8 tools (colors, blue = FN, pink = FP, green = TP)

uniform, where coverage was in general substantially lower than for FPs and FNs (Fig. 4c, top two plots). Closer to their 5′ ends, FNs and TPs were distinguished by their tendencies to overlap exons (Fig. 4c, bottom two plots). Indeed, for all tools, FNs appear to have substantial overlap with exons from other transcript isoforms (Fig. 4a). Overlapping exons may thus be a key obstacle to improving recall of many short-read RI detection tools.

## Persistent introns or called RIs occur in genes with experimentally validated IR

Finally, we searched the literature and third-party resources for independent evidence of persistent introns appearing in the HX1 and iPSC samples studied here. We examined RI presence in six genes (2 in both HX1 and iPSC and 4 in iPSC alone) that have experimentally validated IR from a variety of cell types and tissues (Additional file 1: Table S3) [17,

52–54]. We found that intron retention across these six genes varied substantially by sample (no TP introns were observed in both HX1 and iPSC) (Fig. 5). We also found significant variation between the set of RIs in these genes called by different short-read tools, with no TP introns identified by all tools (Additional file 1: Fig. S14). Interestingly, the *SRSF7* gene, which has previously been shown to exhibit post-transcriptional splicing [24, 55], appeared to be generally enriched for persistent introns.

## Discussion

This work raises fundamental questions regarding how results from short-read RI detection tools should be interpreted. We have taken IR to mean the persistence of an intron in a transcript after processing is complete, in alignment with the biological literature on IR. Short-read RI detection tools are commonly thought to identify such retained introns, with the assumption that poly(A) selection is sufficient to guarantee fully spliced and mature transcripts for sequencing; however, these tools are not inherently designed to distinguish intron retention from contaminating events such as partial transcript processing. This disconnect between how tool developers and tool users employ the same language may be responsible for false assertions in the published literature about which introns are retained. We note, for instance, that the prediction of putative neoepitopes arising from IR [31–35]

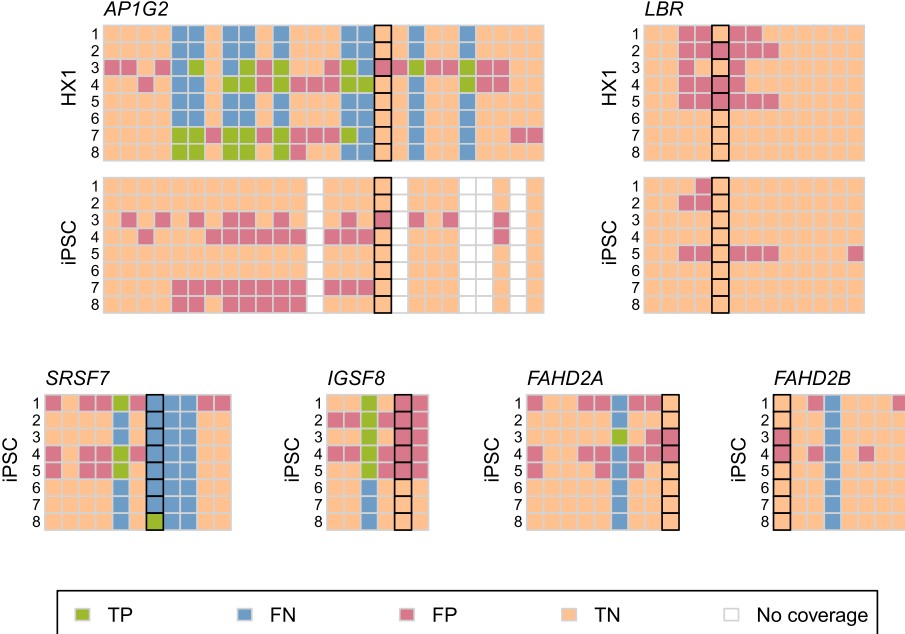

**Fig. 5** Short-read tool performance across six genes with experimentally validated RIs. Comparison of short-read tool called RIs with introns detected in long-read data are shown as a pair of matrices for each of six genes (*AP1G2*, *LBR*, *SRSF7*, *IGSF8*, *FAHD2A*, and *FAHD2B*). The rows in each matrix correspond to the results from each of eight short-read tools (from top to bottom: 1, IntEREst; 2, iREAD; 3, IRFinder-S; 4, superintronic; 5, KMA; 6, rMATS; 7, MAJIQ; 8, SUPPA2) applied to either HX1 (top) or iPSC (bottom) data; columns correspond to all introns found across all annotated transcript isoforms of the indicated gene, ordered by left and then right genomic coordinates. Each cell in the matrix depicts the presence or absence of an intron in short-read and/or long-read data as a TP (green), FN (blue), FP (pink), and TN (peach) assessment; white boxes indicate introns found only in transcripts with < 5 assigned long reads. Black outlines indicate the experimentally validated RI(s) in each gene

requires confidence in the detection of stable, persistent IR with a high likelihood of translation and a low likelihood of undergoing NMD, none of which is assured by short-read RI detection tools.

Limitations of this work include the small number of biological specimens with matched short and deep long read RNA-seq available in the public domain, the lack of replicates of short-read RNA-seq data in this setting, and the limited depth of the long read sequencing data. As a result, we were unable to study the patterns of IR across tissue type and other distinguishing sample characteristics. We confined attention to introns that occur in genes with high coverage in both short and long-read data and did not address either confidence in IR as a function of read depth or systematic biases in gene coverage as a function of sequencing platform. We therefore acknowledge the potential for such biases to introduce the false conclusion in some cases that short read detected IR is spurious when no retained introns are present in corresponding long-read data. Our intron persistence metric represents an improvement over PSI, but it only partially accounts for admixed splicing patterns from different cell types in a mixed-cell sample such as HX1. Like other RI detection studies [15–17, 25, 31, 34, 43], our approach is explicitly linked to annotation (here, GENCODE v35) and therefore reports IR only relative to annotated transcripts, ignoring potential unannotated transcripts. We also did not explore the entanglement of biological and technical effects in the length of persistent introns: shorter introns are more likely to be retained [43, 56, 57], but the length limit of PacBio Iso-Seq reads of up to 10 kilobases means that any molecules with longer persistent introns were not considered in this study. Furthermore, we calculated length-weighted median expression to harmonize short-read tool outputs to long-read intron ranges (Additional file 1: Fig. S15), and this stringent approach may have inflated false negative rates in regions returning high expression magnitudes and variances. Finally, we were only able to evaluate a small subset of the tools available for short read-based RI detection, as many of these tools harbor substantial software implementation and reproducibility challenges.

While there is evidence for cytoplasmic splicing, the phenomenon is rare in many tissues and cell types [9–12]. To better explore intron persistence in future work, it may be worth investigating the differences in detectable intron retention between nuclear and cytoplasmic RNA fractions.

## Conclusions

This is the first study to evaluate the quality of short-read RI detection using short- and long-read RNA-seq data from the same biological specimen. This study also establishes a novel metric capturing the persistence of an intron in a transcript as it is processed using deep long read RNA-seq, and it is the first to interrogate the potential effects of splicing progression during transcript processing and spurious sources of intronic sequence. We find that short-read tools detect IR with poor recall and precision, calling into question the completeness and validity of a large percentage of putatively retained introns called by commonly used methods.

## Methods

### Identification of paired short- and long-read data

Two advanced-search queries were performed on the Sequence Read Archive (SRA) (https://www.ncbi.nlm.nih.gov/sra) on July 13, 2021, and all experiment accession numbers were collected from the query results by downloading the resulting `RunInfo` CSV files. For both searches, the query terms included organism "human," source "transcriptomic," strategy "rna seq," and access "public" with platform varying between the two searches: "pacbio smrt" for the long-read query and "illumina" for the short-read query. The `RunInfo` files were merged and projects with both Illumina and PacBio sequencing performed on the same NCBI biosample were identified. Due to relatively low sequencing depth of PacBio experiments, all projects with fewer than 20 PacBio sequencing runs were eliminated. PacBio experiments conducted on any PacBio platform earlier than RS II were also removed. Two remaining biosamples were chosen as data on which to test RI detection: 1) biosample SAMN07611993, an iPS cell line collected and processed by bioproject PRJNA475610, study SRP098984, with 1 short-read and 27 long-read runs [48], and 2) biosample SAMN04251426 (HX1), a whole blood sample collected and processed by bioproject PRJNA301527, study SRP065930, with 1 short-read and 46 long-read runs [47]. (See the project repository at https://github.com/pdxgx/ri-tests for accession numbers.)

### Long-read data collection, initial processing, and alignment

Raw Iso-Seq RS II data were downloaded from the SRA trace site (https://trace.ncbi.nlm.nih.gov/Traces/sra), via the "Original format" links under the "Data access' tab for each run. These comprised three `.bax.h5` files for both samples, with an additional `.bas.h5` and metadata file for each HX1 run. For both samples, individual runs were processed separately as follows, with differences in handling of the two samples as noted. Subreads were extracted to BAM files from the raw movie files using `bax2bam` (v0.0.8). Circular consensus sequences were extracted using `ccs` (v3.4.0) with `—minPasses  1` set to 1 and `—minPredictedAccuracy 0.90`. Barcodes were removed from CCS reads and samples were demultiplexed with `lima` (v2.2.0). For HX1, the input barcode FASTA files were generated from the Clontech_5p and NEB_Clontech_3p lines from "Example 1" `primer.fasta` (https://github.com/PacificBiosciences/IsoSeq/blob/master/isoseq-deduplication.md). For iPSC, forward and reverse barcode fasta files were downloaded from the study's GitHub page (https://github.com/EichlerLab/isoseq_pipeline/tree/master/data) and merged into a single FASTA file per the `lima` input requirements. Since lima generates an output file for each 5′-3′ primer set, these were merged using `samtools  merge` (`samtools` and `htslib` v1.9). Demultiplexed reads were refined and poly(A) tails removed using `isoseq3 refine` (`isoseq` v3.4.0) to generate full-length non-concatemer (FLNC) reads. FLNC reads were extracted to FASTQ files using `bedtools bamtofastq` (`bedtools` v2.30.0) and aligned to GRCh38 with `minimap2` (v2.20-r1061) using the setting `-ax splice:hq`. Sequence download and processing scripts are available at https://github.com/pdxgx/ri-tests. After processing, the 46 HX1 Iso-Seq runs yielded 945,180 aligned long reads covering 32,837 transcripts of 11,813 genes for HX1, with 13,560 of these transcripts covered by at

least 5 long reads and 4409 unique 5+ read transcripts showing evidence of possible intron retention. In the iPSC sample, we obtained 839,558 aligned long reads covering 31,546 transcripts of 11,992 genes. 12,676 of these transcripts were covered by at least 5 long reads, with 3137 unique 5+ read transcripts showing evidence of possible intron retention.

### Assignment of long reads to transcripts

The long-read alignment files were parsed as follows. GENCODE v.35 [58] annotated transcripts' introns, strand, and start/end positions were extracted from the GENODE v35 GTF file. Then for each aligned long read, spliced-out introns, strand and start/end positions were extracted using `pysam` (v0.16.0.1, using `samtools` v1.10) [59, 60]. A set of possible annotated transcripts was generated, comprising transcripts for which the read's set of introns exactly matched the annotated transcripts' introns sets ("all introns"), or if no such transcripts were found, transcripts for which the read's introns were a subset of the transcripts' intron sets ("skipped splicing"). Then, the best transcript match was chosen from the shortlist of potential matches as the transcript whose length most closely matched the read length. Some reads did not cover the full lengths of their best-matched transcripts, defined by the read alignment start and end position encompassing all introns in the annotated transcript ("full length"); in the case where not all intron coordinates were covered, these were labeled "partial" reads.

### Intron persistence calculation

Intron persistence was calculated only for every transcript that was assigned as the best match for at least 5 reads. We calculated persistence for each intron within these transcripts as the information density of the intron $d_i$ (i.e., the proportion of reads assigned to the transcript that cover intron $i$) multiplied by the mean of the product of three terms across all long reads assigned to that isoform:

1. The *retention*, or presence, $R_{r,i}$ of a given intron $i$ is 1 if the read wholly contains $i$ or 0 if it is absent/spliced out as annotated in read $r$.
2. The *spliced fraction* ($SF_{r,i}$) for a given intron $i$ and read $r$ is defined as

$$SF_{r,i} = \frac{|\{i' \in I : R_{r,i'} = 0\}| + R_{r,i} - 1}{|I| - 1}, \tag{3}$$

   where $I$ is the set of introns spanned by $r$ and $R_{r,i}$ is defined above. This fraction of spliced introns in a read, with the target intron excluded, represents the splicing progression of the read. A mature RNA molecule should tend to have fewer unspliced introns present than an RNA from the same transcript at an earlier point in splicing progression.
3. The scaled *Hamming similarity* ($H_{r,i}$) for a given read $r$ and intron $i$ is defined as the average number of spliced or unspliced introns that match between the target read and other reads assigned to the transcript that have intron $i$ spliced the same as in read $r$, scaled to the number of introns in the isoform:

$$H_{r,i} = \frac{1}{|\{r' \in M^t : R_{r',i} = R_{r,i}\}|}$$
$$\times \sum_{\{r' \in M^t : R_{r',i} = R_{r,i}\}} \frac{|\{i' \in I_{r'} \cap I_r : R_{r',i} = R_{r,i'}\}|}{|I_{r'} \cap I_r|} \,, \tag{4}$$

where $I_r$ is the set of introns spanned by $r$, $I_{r'} \cap I_r$ is the set of introns covered by both $r$ and $r'$, $M^t$ is the set of reads assigned as best matches to the same transcript as $r$ and span the target intron $i$, and $R_{r,i'}$ is as defined above. Any partial reads that are assigned to the transcript as a best match but do not span the target intron are not included in this calculation, and the scaled Hamming similarity between two reads is only calculated for introns covered by both reads. This term accounts for the stochasticity of splicing initiation and progression, since a collection of reads would be more likely to have a dissimilar pattern of unspliced introns if the splicing process remained incomplete.

Persistence $P_{i,t}$ was calculated for each intron $i$ in a given transcript isoform $t$ as information density of the intron $d_i$ times the mean of the product of the three terms above per Eq. 1. Since short reads are not assignable to specific transcripts or isoforms, and certain introns fully or partially recur across multiple transcripts, we set the *intron persistence $P_i$* for a given intron i as the maximum $P_{i,t}$ found for that intron across all transcripts in which it occurs per Eq. 2.

$P_{i,t}$ is entirely determined by retention $R_{r,i}$ (which determines PSI) when all reads cover all introns and have the same splicing patterns as each other; however, for $\sim 11\%$ of data containing noise in sequencing and/or splicing, this PSI contribution likely overestimates intron retention compared with $P_{i,t}$ (Additional file 1: Fig. S16).

### Alignment and BAM generation for short-read data

FASTQs were previously generated by other groups using either Illumina's NextSeq 500 (iPSC [48], run id: SRR6026510) or HiSeq 2000 (HX1 [47], run id: SRR2911306), and files were obtained from the SRA using the `fastq-dump` command from the SRA Toolkit (v2.10.8). A `STAR` (v2.7.6a) [61] index was generated based on the GRCh38 primary assembly genome FASTA (ftp://ftp.ebi.ac.uk/pub/databases/gencode/Gencode_human/release_35/GRCh35.primary_assembly.genome.fa.gz) and GTF (ftp://ftp.ebi.ac.uk/pub/databases/gencode/Gencode_human/release_35/gencode.v35.primary_assembly.annotation.gtf.gz) files from GENCODE v35 [58]. Reads were aligned with STAR to this index using the `-outSAMstrandField intronMotif` option. Primary alignments were retained for reads mapping to multiple genome regions. SAM files output by `STAR` were converted to both sorted and unsorted BAM files using `samtools sort` and `view` (`samtools` v1.3.1), respectively.

Additionally, for use with KMA [37], `bowtie2` (v2.3.4.3) [62] alignments were performed. Alignment statistics may be found in the project repository (https://github.com/pdxgx/ri-tests) and are summarized in Additional file 1: Fig. S17. A FASTA file with intron sequences was generated based on the GRCh38 primary assembly genome FASTA and GTF files from GENCODE version 35 using the `generate_introns.py` script from the KMA package setting 0 bp for the `extension` flag. These intron

sequences were combined with the GRCh38 transcript sequence FASTA file from GEN-CODE version 35 (ftp://ftp.ebi.ac.uk/pub/databases/gencode/Gencode_human/relea se_35/gencode.v35.transcripts.fa.gz), and this combined FASTA was used to create a Bowtie 2 index. Reads were aligned to this index using `bowtie2` according to specifications from KMA [63]. To quantify expression from the Bowtie 2 alignments, eXpress (v1.5.1) [64, 65] was used.

### Selection of target gene subset

Due to variable short- and long-read coverage across the genome, we selected a subset of genes to use for our test dataset to ensure adequate sequencing coverage for RI detection on both platforms. For the short-read data, we chose a coverage cutoff based on the requirements of the short-read RI detection tools used. The two tools with clear coverage requirements are iREAD, which requires coverage of 20 reads across an intron for RI detection, and superintronic, which requires 3 reads per region. Since these are short reads (126 bases for iPSC and 90 for HX1) required over potentially long intronic regions, we chose a median gene-wide coverage (including both intronic and exonic regions) of 2 reads per base, ensuring either consistent coverage across the gene or high coverage in some areas. For the PacBio data, we selected 5 long reads per gene, and a further filter of at least 5 reads assigned as "full length" best matches (see Section 5.3) to a single transcript of the gene, as giving enough information for comparing splicing progression and splicing patterns between reads. We further required these 5 full-length reads' aligned left and right coordinates to fall within 50 bases of the matched transcripts' annotated left and right coordinates. The target gene sets, 1203 genes for iPSC and 1327 for HX1, were chosen from the aligned data, naive to potential RI detection, and then for both short- and long- read data, the gene subset was applied as a filter after running metric calculations or RI detection by short read tools. Within these genes, only transcripts with at least 5 coordinate-matched full-length long reads were studied.

### Intron feature annotation

For the set of target genes, transcripts with at least 5 long reads were selected for analysis. Features of each intron in these transcripts including intron lengths, splice motif sequences, relative transcript position, spliceosome category, and transcript feature overlap properties were extracted as follows. Length was calculated as the difference between the right and left genomic coordinates of the intron ends. Relative position within the transcript is an intron-count normalized fraction where 0 represents the transcript's 5′ end and 1 represents the 3′ end. Splice motifs were assigned to each intron by querying the GRCh38 reference genome with `samtools faidx` (`samtools` v1.10) for the two coordinate positions at each end of the intron, and assigned to one of three canonical motif sequences (GT-AG, GC-AG, and AT-AC, and their reverse complements for − strand genes) or labeled as "other" for noncanonical motifs. Three feature overlap properties were studied: the total number of exons from other transcripts with any overlap of the intron region, the percent of intron bases with at least one overlapping exon from another transcript, and the maximum number of exons overlapping a single base in the intron. These were calculated by extracting all exon coordinates from

the GENCODE v35 annotation file and using an interval tree to query each intron base position against the set of annotated exon coordinates. Spliceosome category was determined from recent U2 and U12 intron annotations [7]. BED files of U2 and U12 introns for GRCh38 were downloaded from the Intron Annotation and Orthology Database (https://introndb.lerner.ccf.org/) on January 25, 2022. Introns were labeled "U2" or "U12" if they only overlapped ranges from one of either spliceosome category, and remaining introns were labeled "other." GC content was calculated for intron sequences by first obtaining intron sequences using the `getSeq()` function from the `BSgenome` (v1.64.0) R package on the `BSgenome.Hsapiens.UCSC.hg38` (v1.4.4) dataset. We then calculated GC frequency with the `letterFrequency()` function from the `Biostrings` (v2.64.0) R package, and we finally obtained the GC fraction by dividing by this frequency by the intron size in bp.

### Selection of short-read RI detection algorithms and identification of likely RIs

We successfully downloaded and ran eight tools for short-read data, including five RI-specific detection tools (superintronic, KMA, IntEREst, iREAD, and IRFinder-S) and three general tools for alternative splicing detection (MAJIQ, rMATS, and SUPPA2). We used a remote server with the CentOS v7 operating system. To run superintronic, KMA, IntEREst, iREAD, MAJIQ, rMATS, and SUPPA2, we used conda virtual environments (see https://github.com/pdxgx/ri-tests). We ran IRFinder-S from a fully self-contained Dropbox image per the tool's instructions (see below). IntEREst and superintronic are provided as R libraries which we ran from interactive R sessions, while iREAD, IRFinder-S, and KMA were run from command line, and a separate R package was used for RI detection for KMA. Outputs from all tools were read into R and harmonized to a single set of intron ranges after applying minimum coverage filters based on both short-read and long-read data. After running tools according to their provided documentation, we consulted literature and documentation on a tool-by-tool basis to devise starting filter criteria based on expression magnitude and other properties. We used these starting criteria to find the subset of most likely RIs, then we modified filter criteria to ensure filtered intron quantities were roughly one order of magnitude lower than unfiltered introns in both iPSC and HX1.

### IR quantification with IntEREst

To run IntEREst (v1.6.2) [38], the `referencePrepare` function from the package was used to generate a reference from the GENCODE v35 primary assembly GTF file [58]. This reference was used along with the sorted STAR BAM alignment from each sample to detect intron retention with the `interest` function, considering all reads and not just those that map to junctions. We used the `interest` function with the `IntRet` setting, which takes into account both intron-spanning and intron-exon junction reads and returns expression as a normalized FPKM. The filter FPKM $\geq$ 3, recommended for iREAD, left > 90% of introns in both samples, so we increased the minimum filter to FPKM $\geq$ 45, and this retained $5038/32544 \approx 15\%$ of introns in HX1 and $6832/21820 \approx 31\%$ of introns in iPSC (Additional file 1: Fig. S10).

### IR quantification with keep me around (KMA)

To run KMA [37], we used devtools to install a patched version of the software which resolves a bug unaddressed by the authors, available at https://github.com/adamtongji/kma. The `read_express` function was used to load expression quantification data output from eXpress, and the `newIntronRetention` function was used to detect intron retention. Returned intron expression was scaled as transcripts per million (TPM). We noted the recommended filters of unique counts $\geq 3$ and TPM $\geq 1$ left just 7.2% of introns in iPSC versus 19% in HX1, so we used a less stringent filter of unique counts $\geq 10$ for both samples, which left $6437/14155 \approx 45\%$ of introns in iPSC and $5089/20484 \approx 25\%$ of introns in HX1 (Additional file 1: Fig. S10).

### IR quantification with iREAD

To run iREAD (v0.8.5) [39], a custom intron BED file was made from the GENCODE v35 primary assembly GTF file using GTFtools (v0.6.9) [66]. The total number of mapped reads in each sorted STAR BAM alignment was determined using samtools, and used as input to the iREAD python script to detect intron retention. Intron expression was returned scaled as FPKM. To identify the most likely RIs, we applied previously published filter recommendations for entropy score ($\geq 0.9$) and junction reads ($\geq 1$). Since there were relatively few introns remaining after applying published filters to the iPSC short-read data ($313/19316 \approx 1.6\%$ vs. $583/7748 \approx 7.5\%$ in HX1), we applied lower filters for FPKM ($\geq 1$ vs. $\geq 3$) and read fragments ($\geq 10$ vs. $\geq 20$) (Additional file 1: Fig. S10).

### IR quantification with superintronic

To run superintronic (v0.99.4) [40], intronic and exonic regions were gathered from the GENCODE v35 primary assembly GTF file [58] using the `collect_parts` function. The `compute_coverage` function was used to compute coverage scores for each sample from sorted STAR BAM alignments, and the `join_parts` function was used to convert these scores to per-feature coverage scores. Intron expression was returned as $\log_2$-scaled coverage, and we identified retained intron ranges as those overlapping long read-normalized ranges with LWM $\geq 3$, per the expressed introns filter described in [40] (Additional file 1: Fig. S10).

### IR quantification with IRFinder-S

We ran IRFinder-S v2.0-beta using the Docker image obtained from https://github.com/RitchieLabIGH/IRFinder. We prepared the IRFinder reference files using the GENCODE v35 genome sequence reference and intron annotations [58]. Our analyses focused on the coverage and IRratio metrics, and the intron expression profile flags returned under warnings. Intron expression was returned as an IRratio, which is similar to PSI, and we identified likely retained introns as having IRratio $\geq 0.5$ without any flags per the methods in [41] (Additional file 1: Fig. S10).

### IR quantification with MAJIQ

We ran the MAJIQ (v2.4) package using our STAR-aligned BAM files and the GENCODE v35 GFF3 genome annotation [45]. We first ran `majiq build` to generate

a file containing all local splice variations (LSVs), followed by `majiq psi` to calculate PSI for each LSV. We then identified the subset of RI LSVs by filtering on terms under the column "`lsv_type`" and available coordinates under the "`ir_coords`" column of the PSI outputs. We identified likely retained introns as having PSI > 0.5, which was similar to the threshold we used for IRFinder-S outputs.

### IR quantification with rMATS

We ran the rMATS (v.4.1.1) Python package, otherwise known as "`rmats-turbo`," using the GENCODE v35 GTF genome annotation file and the sample FASTQ files [44]. This returned a separate series of output files for each alternative splicing event type. We identified the IR splice event outputs as files containing "RI" in the file names, and we extracted the Inclusion Level, which is similar to PSI. We identified likely retained introns as having an Inclusion Level > 0.8 which identified a number of events that was approximately similar to those returned by MAJIQ.

### IR quantification with SUPPA2

We ran the SUPPA2 (v2.3) Python package using the sample FASTQ files [46]. Briefly, SUPPA2 takes as input the isoform-level expression output from the Salmon (v1.9.0) software. We first generated the Salmon index using the GENCODE v35 transcripts FASTA, then we quantified isoform expression using the sample paired FASTQs in TPM units. Next, we generated the splice events annotation with the SUPPA2 `generateEvents` function on the GENCODE v35 GTF, and finally quantified IR events using the `psiPerEvent` function and the Salmon isoform expression output. This outputs the junction-level expression as PSI. We identified likely retained introns as those having PSI > 0.8, as this was similar to the threshold used for rMATS and returned event quantities similar to those from iREAD.

### Harmonization of intron retention metrics across algorithms and runs

Prior to analysis, we harmonized algorithm outputs on intron ranges returned by analysis of available long read runs. We harmonized intron expressions from short read RI detection tools to intron ranges remaining after long reads were uniquely mapped to transcript isoforms. For each short-read RI detection tool, we calculated the region median intron expression value after weighting values on overlapping range lengths (a.k.a. length-weighted medians [LWM]). Calculation of LWMs is shown for an example intron in Additional file 1: Fig. S15. Inter-rater agreement among the output from different short-read algorithms was assessed by Fleiss' kappa [67] using the R package `irr` (v0.84.1.67) [68].

### Determination of truth metric groups from called RIs

For each short-read tool, we used the tool-specific called RIs and the LR persistent introns, as described above, to categorize introns as either true positive (TP, i.e. persistent in long-read and called in short-read), true negative (TN, i.e. neither persistent in long-read nor called in short-read), false positive (FP, i.e. not persistent in long-read, called in short-read), or false negative (FN, i.e. persistent in long-read, not called

in short-read). (Note: We use this language as a convenient description of our results, not to indicate that the long-read persistent introns indicate a full and complete set of true biological retained introns.) We then identified subsets of TP introns identified among at least N tools (e.g. a TP 3+ intron was TP for ≥3 short-read tools, etc.).

### Calculation of performances by intron length bins

We calculated called RI performance metrics across eight short-read tools for a series of overlapping intron length bins. In total, 41 bins were calculated for each sample by sliding 300 bp-wide windows from 0 to 4300 bp lengths at 100 bp intervals. Plots were generated by computing LOESS smooths of the binned performance results.

### Calculation of normalized binned coverages

We evaluated binned intron characteristics across intron truth metric categories for each sample. We assigned introns to truth categories if they were recurrent in that category for ≥ 3 of 8 short-read tools (e.g., an intron that was recurrent TP for 3 tools in iPSC, etc.). We then calculated the log10 median short-read coverage for 1,000 evenly spaced bins per intron for each truth category. We further calculated percent of introns overlapping an exon for each bin by using annotations from the GENCODE v35 GTF. Plots were generated by computing the LOESS smooths of the binned results.

### Comparison of detected RIs with circular RNA

We downloaded a database of human circular RNAs from circbase [50] (http://www.circbase.org/download/hsa_hg19_circRNA.txt), most recently updated in 2017. We extracted all cRNAs labeled with the "intronic" flag in the annotation column and performed a liftover of genomic coordinates for these cRNAs from hg19 to GRCh38 using the UCSC Genome Browser `liftover` tool (https://genome.ucsc.edu/cgi-bin/hgLiftOver). For each sample, we determined the percent of introns overlapping at least one cRNA for the 4+ consensus truth metric groups TP, FP, and FN (e.g., intron was TP in ≥ 4 SR tools, etc.)

### Comparison of detected RIs with validated RIs

In order to test introns in this study against experimentally validated RIs, we identified wet-lab studies in the literature that had first predicted and then validated intron retention. We identified 4 such studies [17, 52–54] that validated a total of 6 RIs in our sets of target genes as defined above (2 and 6 in HX1 and iPSC respectively) (Additional file 1: Table S3). (The above four plus an additional ten studies [9, 30, 69–76] experimentally validated RIs in an additional 2 genes that were found in our target gene set for each sample, but without evidence of IR, and 47 and 43 genes, for HX1 and iPSC respectively, that did not pass our sample coverage thresholds for inclusion in this study.) The validated intron coordinates (Additional file 1: Table S3) were extracted either from the published intron number [17, 52, 53], assuming a count from the gene's 5′ to 3′ end, or via BLAT queries of the target sequence [54]. Adequate intron expression information was available in both samples for the genes *LBR* and *AP1G2*, but only one sample each for remaining genes.

## Supplementary Information

---

Additional file 1. This file includes all supplementary figures and tables referenced in the article.

Additional file 2. This file includes performance benchmarks applying tool-specific filtering criteria.

Additional file 3. Review history.

---

### Acknowledgements
We thank Kasper Hansen, Jeremy Goecks, and Joe Gray for their helpful feedback as this work was being prepared.

### Peer review information

### Review history
The review history is available as Additional file 3.

### Disclaimer
The contents do not represent the views of the U.S. Department of Veterans Affairs or the United States Government.

### Code availability
Scripts to reproduce results contained here are available on GitHub under MIT license [77] and Zenodo [78].

### Authors' contributions
JKD, SKM, RFT, and AN formulated research goals. JKD, SKM, MAW, and RFT wrote and tested computer code. JKD, SKM, and RFT prepared figures. AN and RFT provided computing resources, mentorship, and project oversight. JKD, SKM, RFT, and AN wrote the initial manuscript draft. All authors read and approved the final manuscript.

### Funding
This work was funded in part by VA Career Development Award 1IK2CX002049-01 to R.F.T.

### Availability of data and materials
Our work reused short- and long-read sequencing runs from a human whole blood sample (HX1; SRP065930 on SRA) [47] and a human induced pluripotent stem cell line sample (iPSC; SRP098984 on SRA) [48]. Data and figures generated by scripts are available on GitHub [77] and Zenodo [78].

## Declarations

**Ethics approval and consent to participate**
Not applicable.

**Consent for publication**
Not applicable.

**Competing interests**
The authors declare that they have no competing interests.

**Author details**
[1]Computational Biology Program, Oregon Health & Science University, Portland, OR, USA. [2]Department of Biomedical Engineering, Oregon Health & Science University, Portland, OR, USA. [3]Present Address: Base5 Genomics, Inc., Mountain View, CA, USA. [4]Present Address: Department of Biostatistics, Johns Hopkins Bloomberg School of Public Health, Baltimore, MD, USA. [5]Portland VA Research Foundation, Portland, OR, USA. [6]Present Address: Phase Genomics, Inc., Seattle, WA, USA. [7]Division of Hospital and Specialty Medicine, VA Portland Healthcare System, Portland, OR, USA. [8]Department of Medical Informatics & Clinical Epidemiology, Oregon Health & Science University, Portland, OR, USA. [9]Department of Radiation Medicine, Oregon Health & Science University, Portland, OR, USA. [10]Department of Surgery, Oregon Health & Science University, Portland, OR, USA.

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

## 