## [Additional file 3. Review history. · Genome Biology]

Review History

First round of review

Reviewer 1

Were you able to assess all statistics in the manuscript, including the appropriateness of statistical tests used? There are no statistics in the manuscript.

Comments to author:

This manuscript presents a benchmark of five computational tools to analyze intron retention (IR) using short reads by doing a direct comparison with high-confidence retained introns defined from long read data from matched samples. While the goal of the study is appreciated, there are several major limitations.

1) The selection of software to quantify IR is at the very least incomplete. I can see that the logic of the authors was to select software specifically designed to detect and quantify IR. However, only IRFinder-S is a well-known software in the field. Most researchers use other software that can quantify also other alternative splicing types, in addition to IR. Thus, the benchmark should at least include MISO (which used to be the most used tool), rMATS and MAJIQ (currently the most highly used software), and vast-tools (which implements the method described in one of the most cited IR papers, Braunschweig et al. 2014). Without including these tools, the interest of this benchmark study is very limited.

2) The use of retained introns identified from long reads as a gold standard needs to be better supported. In other words, it was not clear to me whether the set of introns used as a gold standard for the benchmark adequately represent the full set of true retained introns. There are several considerations here. First, it is known that long reads are not perfect and they are not always "full length". This can introduce a bias towards short introns (since transcripts with very long introns are expected to break) and towards the 3' of the gene (for long genes). Also, I am not convinced about the score the authors introduced (intron Persistence). In Fig 2a they show multiple intron examples with different $P_{i,t}$ values. However, most of the examples with $P_{i,t} > 0.2$ actually have very high fractions of retention, so I am not sure how $P_{i,t}$ compares to the more standard percent of retention (or percent-spliced-in, psi) normally used by short-read tools. Related to this, it is not clear to me whether the long reads analyzed and used to calculate $P_{i,t}$ must cover full transcripts (or at least the 3' of transcripts) or they can often represent incomplete transcripts. If the latter, this would mean long read data includes pre-process transcripts and this would not be a fair comparison for short reads, for which polyA selection in library preparations actually works well. This may be accounted for by the authors, but I am not sure based on the illustration of $P_{i,t}$ in Fig. S1. Also, the fraction of intronic nucleotides with overlapping exons seems excessive (Fig. 1b, a median of 90% of the bases overlap exon sequences). Such high overlap goes against the very concept of intronic sequences, so I am not very sure what they are considering as introns here. For instance, is it possible they are taking alternative last exons (which extend through annotated introns) as retained introns? This would be definitely not correct, since these are two different phenomena (I suggest the authors to show some browser screenshots and provide the list of retained intron coordinates). In summary, I suggest that the authors define a new, fairer and simpler, set of true retained introns from long reads. For example: introns for which there are

at least two long reads fully covering the intron plus the upstream and downstream exons (to ensure they are truly retained introns and not APAs, etc.) and that also include the 3' of the transcript/polyA (to ensure they are processed transcripts). Even if these are few cases given the low coverage of the long-read data, these will be a much fairer set of retained introns for comparison.

3) As the authors acknowledge in the Discussion, the study and the conclusions are based solely on the analysis of a single dataset. Therefore bias/errors coming from (unknown) technical limitations of the long and/or short read data in this study cannot be ruled out.

More minor comments:

4) For those cases with sufficient long read coverage, how does $P_{i,t}$ compare to ψ (i.e. simply the percent of transcripts with the intron retained vs those with the intron spliced out)?

5) Fig. 2b: when showing the distribution of retained introns within the transcript, is there a minimum number of introns per transcript required? Transcripts/genes with fewer than e.g. 5 introns should be excluded, as they would bias the distribution.

6) GC content has been associated with IR. It would be interesting to include this feature in the analyses.

7) Fig. 3a: this is probably something I am missing, but shouldn't short read tools perform better for introns with higher $P_{i,t}$ values? If $P_{i,t}$ associates with "true retention", shouldn't it be easier for those tools to pick retained introns?

8) I probably miss this too, but I could not find it in the Methods: what are the sets of TP, FN and FP used in Section 2.4 for each tool?

9) In the Introduction it is stated that there are "several known spliceosomal types, of which the most studied are called U2 and U12". Which ones are the other less studied types?

Reviewer 2

Were you able to assess all statistics in the manuscript, including the appropriateness of statistical tests used? No, I do not feel adequately qualified to assess the statistics.

Comments to author:

This manuscript by David et al. is comparing intron retention (IR) as discovered by long-read PacBio sequencing analysis to IR detected by analysis of sample matched short read data. The main conclusion, that is well supported by the data, is that none of the 5 IR tools that were used in the short read analysis, reliably detect the IR events seen in long reads. The authors also demonstrate that over 50% of IR events were not called by more than one of these tools. Not surprisingly, detection of IR events using these tools were adversely affected by greater intron length and overlap with annotated exons.

I find the methods to be generally appropriate and well described, with inclusion of sufficient details. I also think that this work is novel and represents an advance in the field. I think it will be of significant interest to a broad audience of RNA biologists, since IR is clearly more important than previously appreciated, yet still often poorly documented because of the problems that the authors highlight in this manuscript.

I have a few specific comments:

1) Page 1, bottom: "There is evidence of cytoplasmic splicing" . I find the evidence for this to be rather weak and would prefer that this is changed to something like " although a few publications have presented data to suggest cytoplasmic splicing."

2) Page 2, top: "an intron normally spliced out". I don't agree with this phrasing. There is now ample evidence that IR is important in gene regulation and that IR often occurs as a normal event in normal cells. Please rephrase!

3) Page 2, end of first paragraph: " iR rarely gives rise to a protein product". It is still unclear how rare this is, but there are now many examples of novel protein isoforms in normal cells that are translated from mRNAs with retained introns (see for example some recent "biology" reviews about IR).

4)Page 9, line 36: "Interestingly, the genes SRSF7 and AP1G253 appear to be generally enriched for persistent introns, potentially consistent with post-transcriptional splicing". I do not understand why this would indicate post-transcriptional splicing, rather than introns the remain unspliced throughout the life of these RNAs?

5)Page 10, last line of discussion: "It may be worth exploring the extent to which sequencing only cytoplasmic RNAs focuses attention of fully processed transcripts in future work" This is a bit cryptic and misleading. Since there are many examples of IR mRNAs that reach the cytoplasm and are sometimes translated, both in normal and cancer cells, IR events are also detected in cytoplasmic RNAs. However, it is important to compare nuclear and cytoplasmic RNA fractions to determine which IR RNAs are retained in the nucleus and which are also present in the cytoplasm. Maybe the authors can rephrase this to make this clearer?

Dear Editors:

We appreciate your taking the time to coordinate this review, and we thank the reviewers for the careful attention they paid our manuscript and their excellent feedback. We believe we have addressed all outstanding concerns and that our revised manuscript represents a substantial improvement on our initial submission. Most significantly, we have (1) expanded the selection of software we benchmarked to include tools for general alternative splicing detection (rMATS, MAJIQ, and SUPPA2), not just tools that detect only IR, as suggested by Reviewer #1; (2) increased selection stringency so we consider only those transcripts with the richest data (i.e. full-length robust coverage) in our analysis, addressing Reviewer #1's concerns about fairly defining retained introns from long reads; (3) included a comparison of canonical intron PSI with our novel persistence metric for quantifying IR to illustrate the distinction; and (4) addressed various issues with our language raised by both reviewers. We attach two versions of our new manuscript: one explicitly indicates changes with respect to our initial submission with strikethroughs and text we added in red, while the other is a clean copy. Our point-by-point responses to the reviewers' concerns are below in blue.

Reviewer #1: This manuscript presents a benchmark of five computational tools to analyze intron retention (IR) using short reads by doing a direct comparison with high-confidence retained introns defined from long read data from matched samples. While the goal of the study is appreciated, there are several major limitations.

1) The selection of software to quantify IR is at the very least incomplete. I can see that the logic of the authors was to select software specifically designed to detect and quantify IR. However, only IRFinder-S is a well-known software in the field. Most researchers use other software that can quantify also other alternative splicing types, in addition to IR. Thus, the benchmark should at least include MISO (which used to be the most used tool), rMATS and MAJIQ (currently the most highly used software), and vast-tools (which implements the method described in one of the most cited IR papers, Braunschweig et al. 2014). Without including these tools, the interest of this benchmark study is very limited.

We appreciate the reviewer's insightful suggestion and have implemented rMATS and MAJIQ, including their results along with results from our existing implemented toolset. Figures and text throughout the manuscript have been updated to include these results, accordingly. Despite our best efforts, we were unable to include MISO in our analysis as MISO requires custom database files based on older *hg19* gene annotation despite widespread adoption of *hg38*, and it is therefore not comparable to the results from other tools in our analysis. We were additionally unable to include vast-tools in our analysis as it requires a tool-specific custom database in lieu of a GENCODE GTF file, and is therefore intrinsically not comparable with the results generated from other tools herein, which defined introns with respect to GENCODE v35. Instead, we have included an analysis with SUPPA¹, another widely used tool with 235 citations according to Google Scholar as of July 22, 2022 as well as *hg38* support for intron retention detection.

2) The use of retained introns identified from long reads as a gold standard needs to be better supported. In other words, it was not clear to me whether the set of introns used as a gold standard for the benchmark adequately represent the full set of true retained introns. There are several considerations here. First, it is known that long reads are not perfect and they are not always "full length". This can introduce a bias towards short introns (since transcripts with very long introns are expected to break) and towards the 3' of the gene (for long genes).

¹ Trincado, Juan L., et al. "SUPPA2: fast, accurate, and uncertainty-aware differential splicing analysis across multiple conditions." *Genome Biology* 19.1 (2018): 1-11.

We thank the reviewer for their insightful comment. Indeed, the use of long-read data does not guarantee full-length transcripts, particularly for longer transcripts. We have modified our analysis and discussion to address this critique in the following ways:

1) We have included two additional filters to remove consideration any transcripts that do not have at least 5 best-match assigned reads that both A) span all of the transcript's annotated exons and B) are mapped such that the read's ends align to the expected transcript termini within a 50 base tolerance at either end of the transcript. Note that this length tolerance was chosen empirically according to the observed distribution of transcript length mismatches (Figure R1). All figures and analyses were updated to reflect this higher confidence set of transcripts, with evaluation of short read tools focused on transcripts with full-length long read data.

Figure R1: Distribution of length mismatches between aligned long-read data and left and right transcript termini.

2) We have also softened our discussion of “false positive” results present in short-read but not long-read data, as there may indeed be differences in latent biases between short- and long-read approaches. We are not equipped to conclude that an absence of evidence in this case denotes a bonafide absence.

Also, I am not convinced about the score the authors introduced (intron Persistence). In Fig 2a they show multiple intron examples with different $P_{i,t}$ values. However, most of the examples with $P_{i,t} > 0.2$ actually have very high fractions of retention, so I am not sure how $P_{i,t}$ compares to the more standard percent of retention (or percent-spliced-in, psi) normally used by short-read tools.

We compared the Intron PSI with the corresponding depicted $P_{i,t}$ values from Fig. 2a and find that PSI exhibits poor dynamic range when describing potential intron retention, whereas $P_{i,t}$ is able to account for the spectrum of increasingly noisy splicing/data as shown in Fig. 2a (Figure R2). While indeed the fraction of retention appears to be high in many of the depicted examples, we point the reviewer to the corresponding patterns of noisy splicing which likely undermine confidence in the determination of intron retention.

Figure R2: Comparison of percent-spliced-in (PSI) with persistence values across introns depicted in Figure 2a.

The PSI-like quantity that contributes to our persistence metric is intron retention, $R_{r,i}$. To further address the reviewer's critique, we calculated its contribution to persistence for all long read introns and compared each of these values to corresponding persistence ($P_{i,t}$) values for the matched transcript (see Figure S16). We note there is indeed a high correlation between the two values. In fact, $P_{i,t}$ is entirely determined by its $R_{r,i}$ contribution when all reads cover all introns and have the same splicing pattern as each other. However, for the ~11% of data containing noise in sequencing and/or splicing, the values of this contribution likely overestimate intron retention compared with $P_{i,t}$.

Related to this, it is not

clear to me whether the long reads analyzed and used to calculate $P_{i,t}$ must cover full transcripts (or at least the 3' of transcripts) or they can often represent incomplete transcripts. If the latter, this would mean long read data includes pre-process transcripts and this would not be a fair comparison for short reads, for which polyA selection in library preparations actually works well. This may be accounted for by the authors, but I am not sure based on the illustration of $P_{i,t}$ in Fig. S1.

As noted above, our updated data represents a confident set of approximately full length transcripts, with sufficient read depth to derive meaningful insights related to intron retention for all introns in the transcript.

Also, the fraction of intronic nucleotides with overlapping exons seems excessive (Fig. 1b, a median of 90% of the bases overlap exon sequences). Such high overlap goes against the very concept of intronic sequences, so I am not very sure what they are considering as introns here. For instance, is it possible they are taking alternative last exons (which extend through annotated introns) as retained introns? This would be definitely not correct, since these are two different phenomena (I suggest the authors to show some browser screenshots and provide the list of retained intron coordinates).

We thank the reviewer for raising this concern and would like to clarify this phenomenon: many genes have many potential transcript isoforms that are present in GTF annotation. We assign each considered long read to a specific best-matching transcript isoform from the relevant gene (see Methods), and consider any potential

intron retention within a long read only in the context of that annotated transcript isoform. Reads assigned to a given isoform in annotation are required to cover a subset of the isoform's introns, and we do not consider reads as matched to an isoform if they cover only part of an annotated intron. Definitionally, it is impossible for the intronic sequence to have any overlap with exonic sequences for a given transcript; however, the percent exon overlap term is calculated based on all available transcript isoforms for that gene or any other overlapping genomic element (i.e. antisense transcript). This leads to many situations where an intron in one transcript may be considered an exon (or overlap with one) in a different annotated transcript. Note that we removed ambiguous transcript assignments upfront in our analysis, and reiterate that our analysis is directly based on input annotation.

We show here four independent example IGV browser screenshots at different genes with the final filtered long-read data from iPSC, which we believe illustrate the above phenomenon (Figures R3-R6). The first two examples (Figures R3 and R4) show introns having >50% and <100% overlap with exons from another isoform, and the second two (Figures R5 and R6) show an intron having 100% overlap with an exon from another isoform. These examples illustrate that the final filtered long reads from our pipeline do not represent - nor are the retained introns conflated with - alternative last exons.

Figure R3: Multiple transcript isoforms of RPP30 with aligned long-read data for iPSC for an intron having >50% exonic overlap with a single isoform. Browser tracks show (top) chromosome location and coordinates, (middle) splicing, coverage, and illustrative read subset from the final filtered long read BAM, and (bottom) full GTF annotations for GENCODE v35.

Figure R4: Multiple transcript isoforms of ZSCAN26 with aligned long-read data for iPSC for an intron having >50% exonic overlap with a single isoform. Browser tracks show (top) chromosome location and coordinates, (middle) splicing, coverage, and illustrative read subset from the final filtered long read BAM, and (bottom) full GTF annotations for GENCODE v35.

Figure R5: Multiple transcript isoforms of PPP4R2 with aligned long-read data for iPSC for an intron having 100% exonic overlap with a single isoform. Browser tracks show (top) chromosome location and coordinates, (middle) splicing, coverage, and illustrative read subset from the final filtered long read BAM, and (bottom) full GTF annotations for GENCODE v35.

Figure R6: Multiple transcript isoforms of POLR2J with aligned long-read data for iPSC for an intron having 100% exonic overlap with a single isoform. Browser tracks show (top) chromosome location and coordinates, (middle) splicing, coverage, and illustrative read subset from the final filtered long read BAM, and (bottom) full GTF annotations for GENCODE v35.

In summary, I suggest that the authors define a new, fairer and simpler, set of true retained introns from long reads. For example: introns for which there are at least two long reads fully covering the intron plus the upstream and downstream exons (to ensure they are truly retained introns and not APAs, etc.) and that also include the 3' of the transcript/polyA (to ensure they are processed transcripts). Even if these are few cases given the low coverage of the long-read data, these will be a much fairer set of retained introns for comparison.

We appreciate and have embraced the reviewer's suggestion whole-heartedly, and have refined our set of long read data to only consider those transcripts with the richest data (i.e. full-length robust coverage). We have updated our methods, results and text throughout accordingly (please see revised manuscript with tracked changes).

3) As the authors acknowledge in the Discussion, the study and the conclusions are based solely on the analysis of a single dataset. Therefore bias/errors coming from (unknown) technical limitations of the long and/or short read data in this study cannot be ruled out.

We would like to clarify that the two samples we have analyzed represent two entirely independent sets of sequencing experiments from two different laboratories at different institutions using different methods on different specimens, and not a single dataset as the reviewer suggests. Note that we sought to include additional experiments in our analysis, however no additional data met our stringent criteria for inclusion. We commented on this in our discussion: "Limitations of this work include the small number of biological specimens with matched short and deep long read RNA-seq available in the public domain, the lack of replicates of short-read RNA-seq data in this setting, and the limited depth of the long read sequencing data. As a result, we were unable to study the patterns of IR across tissue type and other distinguishing sample characteristics."

More minor comments:

4) For those cases with sufficient long read coverage, how does $P_{i,t}$ compare to ψ (i.e. simply the percent of transcripts with the intron retained vs those with the intron spliced out)?

Please see Figure R2 and above discussion regarding ψ v. $P_{i,t}$. We note that $P_{i,t}$ is generally consistent with ψ ($R=0.96$) but accounts more robustly for splicing and/or sequencing noise.

5) Fig. 2b: when showing the distribution of retained introns within the transcript, is there a minimum number of introns per transcript required? Transcripts/genes with fewer than e.g. 5 introns should be excluded, as they would bias the distribution.

We have amended our methods and results as discussed above - transcripts with fewer than 5 mapped full-length long-reads were excluded from analysis.

6) GC content has been associated with IR. It would be interesting to include this feature in the analyses.

We thank the reviewer for this suggestion, and we now include an assessment of intronic GC content in Figures 2b and 4a. We did not, however, observe an obvious association between GC content and IR.

7) Fig. 3a: this is probably something I am missing, but shouldn't short read tools perform better for introns with higher $P_{i,t}$ values? If $P_{i,t}$ associates with "true retention", shouldn't it be easier for those tools to pick retained introns?

The reviewer raises an insightful critique, and we have puzzled over this as well. With the introduction of rMATS, MAJIQ, and SUPPA2 we note that this phenomenon is no longer uniform. Indeed, for these additional tools we note increasing recall with higher $P_{i,t}$ thresholds, as the reviewer anticipates. However, we would like to reiterate that the higher $P_{i,t}$ along the x-axis of Figure 3a represents $P_{i,t}$ thresholds below which an intron is excluded from being considered retained; as such, lower $P_{i,t}$ threshold values still describe introns with high $P_{i,t}$, these data are just increasingly admixed with other introns having lower $P_{i,t}$. As in Figure R2, we note that lower $P_{i,t}$ values may still connote detection of significant quantities of intronic sequences, which should indeed be detectable by short-read tools; but neither short-read tools, nor the ψ itself accounts for additional information conferred by long-read data penalizing noisy splicing patterns from being considered bonafide IR – short-read data is ill-equipped to pick up on these nuances. Finally, we note that with increasing $P_{i,t}$ thresholds, the actual quantity of introns considered decreases substantially, such that we suspect that measured

precision, recall, and F1 scores are substantially more sensitive to small changes in the quantity of correct v. incorrect classifications at these highest thresholds.

8) I probably miss this too, but I could not find it in the Methods: what are the sets of TP, FN and FP used in Section 2.4 for each tool?

We have clarified these category definitions in a new Methods section and added references to this where relevant in the text.

9) In the Introduction it is stated that there are "several known spliceosomal types, of which the most studied are called U2 and U12". Which ones are the other less studied types?

We thank the reviewer for pointing out this grammatical inaccuracy, which has now been fixed in the text; we had intended to say U2 and U12 alone.

Reviewer #2: This manuscript by David et al. is comparing intron retention (IR) as discovered by long-read PacBio sequencing analysis to IR detected by analysis of sample matched short read data. The main conclusion, that is well supported by the data, is that none of the 5 IR tools that were used in the short read analysis, reliably detect the IR events seen in long reads. The authors also demonstrate that over 50% of IR events were not called by more than one of these tools. Not surprisingly, detection of IR events using these tools were adversely affected by greater intron length and overlap with annotated exons.

I find the methods to be generally appropriate and well described, with inclusion of sufficient details. I also think that this work is novel and represents an advance in the field. I think it will be of significant interest to a broad audience of RNA biologists, since IR is clearly more important than previously appreciated, yet still often poorly documented because of the problems that the authors highlight in this manuscript.

I have a few specific comments:

1) Page 1, bottom: "There is evidence of cytoplasmic splicing" . I find the evidence for this to be rather weak and would prefer that this is changed to something like " although a few publications have presented data to suggest cytoplasmic splicing."

We have amended the text to soften this language per the reviewer's suggestion.

2) Page 2, top: "an intron normally spliced out". I don't agree with this phrasing. There is now ample evidence that IR is important in gene regulation and that IR often occurs as a normal event in normal cells. Please rephrase!

We agree with the reviewer and note that we previously included a description of the relevance of IR for e.g. gene regulation in the introduction. We have now additionally removed the phrase highlighted by the reviewer and amended the text to rephrase according to the reviewer's concern.

3) Page 2, end of first paragraph: " iR rarely gives rise to a protein product". It is still unclear how rare this is, but there are now many examples of novel protein isoforms in normal cells that are translated from mRNAs with retained introns (see for example some recent "biology" reviews about IR).

We agree with the reviewer, and note that the second half of the quoted sentence originally read as follows: "but novel peptides derived from transcripts with retained introns (RIs) are increasingly being studied in

disease contexts such as cancer [31–35]”. We have amended this text to highlight the potential for IR to generate peptides in normal cells.

4)Page 9, line 36: "Interestingly, the genes SRSF7 and AP1G253 appear to be generally enriched for persistent introns, potentially consistent with post-transcriptional splicing". I do not understand why this would indicate post-transcriptional splicing, rather than introns the remain unspliced throughout the life of these RNAs?

Post-transcriptional splicing is indeed a potential explanation for this phenomenon, but the reviewer appropriately suggests a plausible alternative explanation. We have therefore modified the text to remove this explicit attribution.

5)Page 10, last line of discussion: "It may be worth exploring the extent to which sequencing only cytoplasmic RNAs focuses attention of fully processed transcripts in future work" This is a bit cryptic and misleading. Since there are many examples of IR mRNAs that reach the cytoplasm and are sometimes translated, both in normal and cancer cells, IR events are also detected in cytoplasmic RNAs. However, it is important to compare nuclear and cytoplasmic RNA fractions to determine which IR RNAs are retained in the nucleus and which are also present in the cytoplasm. Maybe the authors can rephrase this to make this clearer?

We agree with the reviewer and have endeavored to modify the language in the discussion accordingly.

Second round of review

Reviewer 1

The authors have carefully addressed the concerns I have raised.

I am still a bit puzzled by the relationship between PSI and "intron persistence" (Figure R2). In particular, I agree with the authors (based on Figure R2) that the dynamic range of PSI is low for highly retained introns, although to some extent this seems biologically irrelevant once $PSI > 0.9$. Moreover, the same could be argued for Intron persistence for lowly retained introns, which, arguably, will be the most abundant in a physiological context (even if caused by splicing noise). For this reason, I would not be sure how to interpret, e.g. a Pt of 0.1. I guess a way to address this is to do RT-PCR assays and compare the results with the two metrics. It is true that many validations of IR do not faithfully recapitulate the expected high PSI values.

In any case, I do not have any specific suggestion here. I just recommend the authors to give this an additional thought, since researchers in the splicing field are used to use PSIs.

Authors Response

Point-by-point responses to the reviewers' comments:

Comment:

The authors have carefully addressed the concerns I have raised.

I am still a bit puzzled by the relationship between PSI and "intron persistence" (Figure R2). In particular, I agree with the authors (based on Figure R2) that the dynamic range of PSI is low for highly retained introns, although to some extent this seems biologically irrelevant once $PSI > 0.9$. Moreover, the same could be argued for Intron persistence for lowly retained introns, which, arguably, will be the most abundant in a physiological context (even if caused by splicing noise). For this reason, I would not be sure how to interpret, e.g. a Pt of 0.1. I guess a way to address this is to do RT-PCR assays and compare the results with the two metrics. It is true that many validations of IR do not faithfully recapitulate the expected high PSI values.

In any case, I do not have any specific suggestion here. I just recommend the authors to give this an additional thought, since researchers in the splicing field are used to use PSIs.

Response

We have slightly modified a statement in Discussion given Reviewer #1's feedback, now writing, "Our intron persistence metric represents an improvement over PSI, but it only partially accounts for admixed splicing patterns from different cell types in a mixed-cell sample such as HX1."